# Ocular Drug Delivery: Advancements and Innovations

**DOI:** 10.3390/pharmaceutics14091931

**Published:** 2022-09-13

**Authors:** Bo Tian, Evan Bilsbury, Sean Doherty, Sean Teebagy, Emma Wood, Wenqi Su, Guangping Gao, Haijiang Lin

**Affiliations:** 1Department of Ophthalmology and Visual Sciences, University of Massachusetts Chan Medical School, Worcester, MA 01655, USA; 2Horae Gene Therapy Center, University of Massachusetts Chan Medical School, Worcester, MA 01605, USA; 3Department of Microbiology and Physiological Systems, University of Massachusetts Chan Medical School, Worcester, MA 01605, USA; 4Viral Vector Core, University of Massachusetts Chan Medical School, Worcester, MA 01605, USA; 5Li Weibo Institute for Rare Diseases Research, University of Massachusetts Chan Medical School, Worcester, MA 01605, USA

**Keywords:** ocular drug delivery, gene therapy, adeno-associated virus, non-viral vectors, medication carriers, administration routes

## Abstract

Ocular drug delivery has been significantly advanced for not only pharmaceutical compounds, such as steroids, nonsteroidal anti-inflammatory drugs, immune modulators, antibiotics, and so forth, but also for the rapidly progressed gene therapy products. For conventional non-gene therapy drugs, appropriate surgical approaches and releasing systems are the main deliberation to achieve adequate treatment outcomes, whereas the scope of “drug delivery” for gene therapy drugs further expands to transgene construct optimization, vector selection, and vector engineering. The eye is the particularly well-suited organ as the gene therapy target, owing to multiple advantages. In this review, we will delve into three main aspects of ocular drug delivery for both conventional drugs and adeno-associated virus (AAV)-based gene therapy products: (1) the development of AAV vector systems for ocular gene therapy, (2) the innovative carriers of medication, and (3) administration routes progression.

## 1. Introduction

The anatomical and physiological barriers of the eye render this organ largely impervious to external factors, protecting against pathogen entry while simultaneously impeding drug permeation. The anterior and posterior segments of the eye both present their own unique challenges for drug delivery, as illustrated in Figure 1. The anterior chamber comprises the cornea, aqueous humor, and lens [1]. In the anterior of the eye, the blood–aqueous barrier, consisting of the iris/ciliary blood vessels and nonpigmented ciliary epithelium, limits access to the anterior of the eye and hampers therapeutic entry to the intraocular environment [2]. In the posterior of the eye, the blood retinal barrier, comprised of the retinal capillary endothelial cells and retinal pigment epithelial cells, prevents therapeutic agents from entering the posterior segment from the bloodstream [3]. Due to the high compartmentalization and broad range of ophthalmic pathology, the drug formulation, delivery devices, and administration routes need to be tailored to therapeutic strategies, overcoming the pharmacokinetic limitations, meanwhile achieving the adequate bioavailability. Ocular pharmaceutical sciences, therefore, have been advancing the delivery system for prolonging retention and reducing elimination to enhance the drug treatment efficiency.

As a novel form of “drug”, gene therapy that produces therapeutic biological agents within specifically targeted cells has shown significant clinical therapeutic progression in the past decade. Ophthalmic gene therapy has been at the forefront of gene therapy research; the compartmentalization, privileged immunity, physical accessibility, and the post-mitotic status of the cells render the eye an extremely attractive organ for adeno-associated virus (AAV)-based gene therapy, a leading platform of ocular gene therapy. Since the advent of the first FDA-approved gene therapy, Luxturna^TM^, for the treatment of type 2 Leber congenital amaurosis (LCA) [4], ophthalmic gene therapy research has blossomed. Despite of the initial exploration for inherited disorders in the field, applications of gene therapy are expanding beyond rare inherited diseases to acquired disorders with higher prevalence, including age-related macular degeneration (AMD), diabetic retinopathy (DR), glaucoma, and corneal diseases [5]. There are over 40 clinical trials ongoing for AAV-based ocular gene therapy products, aiming to treat both inherited and non-inherited eye diseases, listed in Table 1. Hence, as the viral vectors, the AAV gene therapy products will be mainly reviewed. Ocular gene therapy to date has mostly involved viruses as carriers of the gene, although the research in non-viral vectors has also advanced significantly to overcome some of the unique limitations of viral vectors, such as immunogenicity and packaging limitation. The development of non-viral vectors will be also discussed.

Overall, this review will discuss the advancements of ocular drug delivery for both conventional pharmaceutical compounds and AAV-mediated gene therapy products, including the development and advancement of AAV vectors for ocular gene therapy, drug delivery systems, and technological advances in ocular drug administration.

## 2. Ocular Gene Therapy

Driven by the advances in viral vector technology and discovery in genetic basis for ocular disorders, gene therapies in this field have been surging in past decades, culminated in the FDA approval of Luxturna™ in 2017, the first ocular gene therapy product. This milestone event further boosts the research and clinical interests in gene therapy for a broader scope of ophthalmic diseases. There are more than 350 hereditary ocular diseases, including retinitis pigmentosa, choroideremia, Stargardt disease, Leber’s congenital amaurosis (LCA), involving a wide diversity of genetic loci [6,7]. With multiple clinical trials underway, gene therapy is acknowledged as having potential in the treatment of the inherited eye diseases. Further, the gene therapy approach is also being developed and expanded to conditions not associated with a single genetic defect, such as corneal and retinal vascular diseases in the retina and cornea, or age-related macular degeneration (AMD) [8,9]. AAV, as the most common viral vector for ocular gene therapy, will be reviewed in this article.

The vector engineering, transgene packaging has greatly contributed to transgene expression improvement and phenotypic rescue after intraocular delivery. Despite the success of viral vectors in ocular gene therapy, there are concerns in viral genome heterogeneity during manufacture, packaging capacity, and immunological reactions. Non-viral vectors are being developed to obviate these limitations in gene therapy. We here review the growing body of literature and clinical trials, covering the gene therapies for inherited and acquired eye diseases, as well as the advances in the viral and non-viral vectors in this field. 

### 2.1. AAV-Based Ocular Gene Therapy

#### 2.1.1. Adeno-Associated Viruses (AAVs) 

Adeno-associated viruses (AAVs) were originally discovered in the mid-1960s as satellite viruses associated with adenoviruses, hence the name ‘adeno-associated’ [10]. They are small (~26 nm diameter), icosahedral-structured capsid, non-enveloped, single-stranded DNA, non-pathogenic viruses [11]. The viral life cycle of AAV cannot initiate or complete without genes provided by helper viruses, such as adenoviruses, herpesvirus, papillomaviruses, and baculovirus [10]. The viral genome of AAV is linear, containing four known open reading frames (ORFs) for viral genes, flanked by 145 bp inverted terminal repeats (ITRs) at both 5′ and 3′ ends (Figure 2). To package the vector construct, AAV was engineered to generate recombinant AAV (rAAV). The rAAV was created by removing all viral genomic components other than the ITRs, resulting in a packaging capacity up to 4.9 kb for therapeutic applications (Figure 2) [12]. The rep and cap are provided in trans during the production of the AAV vector. Due to its efficiency, low immunogenicity, and lack of pathogenicity, rAAV is currently the leading platform for gene delivery to treat inherited and non-inherited human diseases in both preclinical and clinical settings. 

In 2017, Luxturna^TM^ was the first gene therapy to be approved by the US Food and Drug Administration (FDA). It is a rAAV2-based platform that delivers an RPE65 expression cassette to treat type 2 Leber Congenital Amaurosis (LCA), a recessive monogenetic retinal dystrophy caused by biallelic pathogenic mutation in the *RPE65* gene [13]. With the clinical success for LCA2 patients, AAV-based gene transfer is now being explored clinically for other forms of hereditary retinal diseases, including choroideremia, Leber hereditary optic neuropathy (LHON; NCT01267422 and NCT02161380), Stargardt disease (NCT01367444), X-linked retinoschisis (NCT02317887, NCT02416622), and X-linked retinitis pigmentosa (NCT04671433, NCT03116113, NCT04517149, NCT04850118). All current clinical trials targeting inherited retinal diseases are listed in Table 1. The preclinical and clinical trials are flourishing though, the complexity and challenges of retinal gene therapy have become clear this past year with pivotal clinical trials for RPGR (X-linked retinitis pigmentosa GTPase regulator), retinitis pigmentosa (NCT03116113), and choroideremia (NCT03496012) failing to meet their respective primary endpoints. 

AAV-directed gene therapy also offers novel treatment regimens for tackling a range of common acquired ocular disorders. Although the pathologies of these disorders are more complex than monogenetic inherited diseases, the identification of specific therapeutic targets and the lack of need for repeat dosing has generated much interest for the potential of AAV-mediated gene therapy to treat diseases, such as age-related macular degeneration (AMD) [14], diabetic retinopathy (DR) [15], and glaucoma [16]. These are diseases that necessitate repeated administration of therapeutic agents, a problem that could be overcome by a one-time dose of an AAV-based gene therapy product.

#### 2.1.2. Age-Related Macular Degeneration (AMD)

Age-related macular degeneration (AMD) is the leading cause of irreversible central vision loss in individuals over 65 years of age, affecting more than 1.8 million Americans [17]. There are two forms of AMD, a ‘dry’, non-neovascular form and a ‘wet’, neovascular form. Although advances in cancer biology have given the field potent neovascularization inhibitors, the underlying pathology of the dry form is still poorly understood [17]. Recent advances have identified the complement cascade as playing a role in the development of drusen, which forms over the macula, leading to central-vision loss [18,19]. Although there is still no cure for dry AMD, several AAV-based pharmaceuticals are currently revolutionizing AMD treatment by targeting components of the complement cascade. An AAV-mediated soluble form of CD59 (HMR59, AAV.CAG.sCD59) was developed to block complement in the membrane attack complex (MAC) by Hemera Biosciences Inc. The trial of HMR-1001 evaluated 17 eyes of 17 subjects with geographic atrophy at the advanced stage of non-neovascular AMD. Participants received a single intravitreal dose of HMR59 in phase I study to evaluate the safety and tolerability the of HMR-1001 (NCT03144999). The subsequent phase I trial, HMR-1002, has evaluated the safety and efficacy of HMR59 (AAV2.CAG.sCD59) among 25 patients with new-onset neovascular AMD. Participants received two doses (3.56 × 10^11^ vg and 1.071 × 10^12^ vg; vg: viral genomes) of HMR59 by intravitreal injection 7 days after a single intravitreal injection of anti-VEGF (day 0). A seven-day tapering dose of oral prednisone was started at day 30 by all participants (NCT03585556). Similar trials targeting components of the complement cascade are being carried out by Iveric Bio. These trials are summarized in Table 1. 

Anti-VEGF (vascular endothelial growth factor) therapy blocks the activity of VEGF and has demonstrated significant benefits to neovascular AMD patients [20,21]. However, anti-VEGF therapy is hampered by the short half-life of the protein drug and, therefore, demands frequent intravitreal injections, which places a high burden on both the patient and provider [22]. AAV-based gene therapy has the potential to overcome this issue by mediating a sustained expression of anti-VEGF molecules. AAV-based anti-VEGF gene therapy products have been developed for the treatment of neovascular AMD and evaluated in clinical trials. RGX-314 (RegenxBio Inc., Rockville, MD, USA) is an AAV8-mediated monoclonal anti-VEGF Fab, designed to neutralize VEGF activity. In March 2017, a phase 1/2 trial was launched to evaluate the safety and tolerability of a, single subretinal injection of RGX-314 with five dosing cohorts: 3 × 10^9^, 1 × 10^10^, 6 × 10^10^, 1.6 × 10^11^ and 2.5 × 10^11^ GC/eye (GC: genome copy) (NCT03066258). Overall, a dose-dependent increase in RGX-314 protein expression was observed across all five dosing cohorts at 1 year, which was stable over 2 years in cohort 3, and over 1 year in cohorts 4 and 5. This implies a significant reduction in injection frequency and burden for patients with neovascular AMD, who traditionally receive regular monthly anti-VEGF injections. In September 2020, RegenxBio initiated a phase 2 trial (NCT04514653) to evaluate the efficacy, safety, and tolerability of RGX-314 delivered to suprachoroidal space (SCS) using the suprachoroidal microinjector. SCS injections are a less invasive procedure than intravitreal injections that could potentially offer a higher transduction efficiency of the RPE and photoreceptor cell layer [23]. The SC administration route will be separately discussed in Section 4.

Additionally, ADVM-022, an AAV2.7m8 encoding aflibercept, was developed to express the humanized recombinant protein that is constructed by fusing binding domains from VEGF receptor-1 and receptor-2. The phase 1 clinical trial (OPTIC; NCT03748784) of ADVM-022 demonstrates the potential of ADVM-022 to reduce the injection frequency. A single of intravitreal delivered ADVM-022 (6 × 10^11^ vg/eye) combined with a single dose of oral steroids resulted in a maintained therapeutic effect for over 15 months. A complete list of current clinical trials using AAVs to treat AMD is available in Table 1.

#### 2.1.3. Diabetic Retinopathy

Diabetic retinopathy (DR) is one of the most common complications of diabetes mellitus (DM) and is one of the leading causes of preventable vision loss in the developed world [24]. The onset of diabetic retinopathy is characterized by morphologic alteration of microvessels, with selective loss of pericytes, thickening of the basement membrane, and loss of inter-endothelial tight junctions. These changes lead to increased vascular permeability, capillary occlusion, and microaneurysms, eventually leading to proliferative diabetic retinopathy (PDR) [25]. The increase in vascular permeability, similarly, leads macular edema to neovascular AMD. Due to the overlapping treatment targets between AMD and PDR there has been much concurrent progress in the development of AAV-based gene therapy products targeting these two diseases. Pivotal studies, such as RISE/RIDE (NCT00473382/NCT00473330) and VIVID/VISTA (NCT01331681, NCT01363440) have established the critical role of anti-VEGF intravitreal injections in the treatment of diabetic macular edema (DME) by exhibiting better visual outcomes than focal macular laser photocoagulation. Clinical trials for AAV-based anti-VEGF products, including RGX-314 and ADVM-022, for the treatment of PDR and DME have run alongside the trials for AMD. The similarities and differences between these respective types of clinical trials are displayed in Table 1.

#### 2.1.4. Glaucoma

As a neurodegenerative disease, damaging retinal ganglion cells and their axons, glaucoma is the leading cause of irreversible blindness worldwide and comprises a group of eye diseases that can cause vision loss by damaging the optic nerve [26]. Elevated intraocular pressure (IOP) is a major risk factor associated with optic neuropathy [27]. By the year 2040, glaucoma is projected to affect 111.8 million people worldwide [28]. Despite multiple mechanisms that are involved in glaucoma development and progression, lowering the IOP is currently the main treatment regimen, including daily eye drops or surgery [29]. Patient’s compliance with topical drop instillation is notoriously poor due to the frequency of the eye drops that must be strictly followed to achieve efficacious control of the IOP [30]. Laser treatment and invasive surgery, on the other hand, are not always effective in IOP control. As a group of non-monogenic diseases with unknown etiology, glaucoma is amenable to gene therapy, reducing the treatment burden of glaucoma patients. IOP is maintained physiologically by the trabecular meshwork (TM) in the anterior chamber of the eye [31]. It has been shown in pre-clinical studies using animal models that single-stranded DNA AAVS (ssAAVs) are unable to transfect the TM, despite their high-degree of effectiveness at transfecting cells in the posterior compartment. Instead, double-stranded self-complementary DNA AAVs (scAAVs) have been shown to preferentially transfect the TM when injected into the anterior compartment [32]. A single intracameral injection of scAAV2.CMV.GFP in non-human primates (NHPs) was shown to result in fluorescence in the TM for 2 years post-injection [33]. Recently, Rodriguez-Estevez et al. examined the transduction efficiency and cellular entry of seven serotypes of AAV in both the ss and ds forms and found that the ds form consistently had a 10x higher transduction efficiency [34]. 

Additionally, to prevent glaucomatous neurodegeneration, the neuroprotection approach has drawn substantial attention. Recent work has focused on the delivery of neuroprotective factors to the retina to protect against the optic nerve and retinal ganglion cell damage that occurs in the setting of elevated IOP [35]. Current pre-clinical trials exploring the potential of AAV-mediated gene therapy to treat glaucoma are listed in Table 2.

#### 2.1.5. Corneal Diseases 

Corneal disease is the fourth leading cause of blindness worldwide after cataracts, glaucoma, and age-related macular degeneration [49]. Corneal diseases encompass a range of pathologies, including genetic diseases, mechanical injuries, chemical burns, allergic reactions, and infections [8]. Due to the immune-privileged nature of the cornea and the minimal invasive routes required for vector administration, significant progression in the arena of gene therapy for both inherited and acquired corneal disorders has been achieved over the decades. Although these therapies have not yet advanced to the stage of clinical trials, there is great as-yet-unexplored potential for AAVs in the treatment of corneal diseases.

The success of AAV-based gene therapy in treating inherited retinal diseases has logically pointed to inherited corneal diseases as a potential new frontier. Corneal dystrophies (CDs) that display a Mendelian inheritance pattern due to their monogenetic nature are attractive candidates for gene therapy [50,51]. The size limitation of the cargo for AAV-based gene therapy means that only genes with sizes ranging from 0.9 kb to 2.7 kb can be delivered using AAV. Recessive CDs, such as macular corneal dystrophy (MCD), gelatinous drop-like corneal dystrophy (GDCD), and congenital hereditary endothelial dystrophy II (CHED II), are attractive targets for AAV-based gene therapy because the open-reading frames (ORF) of the three genes involved (*CHST6*, *M1S1*, and *SLC4A11*) range from 0.9 to 2.7 kb, the ideal size for an AAV-based gene complementation approach [52]. All CDs with identified gene mutations are listed in Table 3.

In addition to genetic diseases of the cornea, AAV-mediated gene therapy has a role in the treatment of acquired diseases, such as corneal neovascularization (CoNV), corneal haze, and corneal fibrosis, which are all acquired pathologies of the cornea that AAV-mediated gene therapy displays high-potential to treat, efficaciously and safely. AAV-based approaches to reducing corneal inflammation, CoNV, and fibrosis have been shown to improve outcomes following mechanical or chemical insult to the cornea. We recently showed that a single intrastromal injection of AAV8-KH902 (AAV-vectored conbercept, a VEGF-inhibitor) has been shown to effectively inhibit corneal neovascularization (CoNV) in the alkali burn model of CoNV for upwards of 3 months [40]. In contrast, a single intrastromal injection of conbercept was only efficacious in inhibiting CoNV for 10–14 days [40]. Other strategies are outlined in Table 2.

Similar strategies have been employed for the treatment of infectious keratitis. With the exception of herpes simplex virus (HSV)-mediated keratitis and bacterial keratitis where anti-microbial strategies may be pursued, infectious keratitis treatment focuses on preventing and limiting inflammation, which can lead to corneal haze. For this reason, AAV-mediated gene therapy to reduce inflammation has been pursued; recent strategies are listed and briefly described in Table 2.

Despite aggressive medical treatment, the cornea can still become irreversibly damaged or scarred. When this occurs, the only viable option available to restore vision is corneal transplantation. Although this operation is generally highly successful, in ‘high-risk’ corneas, which are significantly inflamed or show a high degree of neovascularization, the success rate may be as low as 30% over 10 years [53]. Recently, VEGF-inhibitors delivered via intrastromal injection have been used clinically to lower the risk of rejection [54]. However, these agents must be delivered frequently, which introduces the risk of corneal scarring, leading to vision loss. Therefore, to facilitate the process of corneal transplantation and prevent scarring from repeated injections, AAV-mediated anti-VEGF therapeutics may be used to reduce CoNV and improve graft survival rate. We recently showed that AAV8-KH902 is effective in reducing corneal transplant rejection in a rat model of high-risk keratoplasty [45], achieving superior allograft survival rates and sustained anti-VEGF expression, compared to the traditional administration of anti-VEGF agents. 

A summary of pre-clinical studies that use AAVs as vectors for ocular gene therapy in the cornea is provided in Table 2.

### 2.2. Organic Nanoparticles (NPs) 

Significant research efforts have been directed towards the development of non-viral DNA delivery systems. Although there has been much focus on using replication-deficient viral vectors to deliver gene therapy products to the eye, non-viral vectors offer some unique advantages over viral vectors. Notably, non-viral vectors have significantly lower immunogenicity, limited size constraints, and ease of large-scale production [55]. Conversely, non-viral vectors lack the cell-type specific tropism of viral vectors and are generally less effective at inducing sustained transgene expression. The lack of tropism makes the method by which cells uptake non-viral vectors extremely important. Target cells may engulf the vector by endocytosis or phagocytosis. Notably, while photoreceptors are predominantly endocytic cells, RPE cells are mainly phagocytic, due to their role in phagocytizing photoreceptor outer segments in vivo [56]. The bioavailability of gene therapy products may be enhanced by non-viral vectors due to the presence of P-gp and multi-drug efflux pumps, which limit the bioavailability of conventional drugs [57,58]. Additionally, functionalization of the outer membrane, as well as the addition of nanostructured lipid carriers (NLCs) to the gene therapy product, serve to enhance tissue/cell-specific tropism and enhance transduction efficiency.

Non-viral NPs can be broadly classified into two groups: organic and inorganic. Inorganic NPs, such as gold NPs, sliver NPs, silica NPs, magnetic NPs, and nanoceria are also being explored in this field, while the discussing all of them would be space prohibitive and reviews illustrating them have been published [59]. Organic NPs include liposomes, niosomes, solid lipid NPs (SLNs), nanostructured lipid carriers (NLCs), polymer/peptide NPs, dendrimers, and nanoemulsions [60]. The properties of these vectors, as well as examples of their uses and advantages and disadvantages, will be discussed below. 

#### 2.2.1. Solid Lipid Nanoparticles (SLNs) and Nanostructured Lipid Carriers (NLCs)

Solid lipid nanoparticles (SLNs) and nanostructured lipid carriers (NLCs) are modified liposomes that have been engineered to be more suited for gene/drug delivery than simple phospholipid bilayers (Figure 3). Although liposomes typically have an aqueous core surrounded by a lipid bilayer, SLNs feature a solid-lipid crystal core that is stabilized by surfactants, which act as emulsifiers. Although any lipids could theoretically be used as components of the core and outer bilayer, physiologic lipids are preferred because they do not confer any toxicity or inflammatory response. Importantly, these particles are able to undergo autoclave sterilization while retaining their functionality [61].

SLNs are assembled electrostatically from cationic liposomes and anionic protamine-DNA complexes that form liposome protamine/DNA lipoplexes (LPDs). Protamine temporarily replaces histones in the process of spermatogenesis and efficiently condenses DNA, making it very useful in preventing the replication of pathologic cells and, therefore, the spread of disease. Despite their many advantages, SLNs are limited in their drug-loading capacity and can erroneously expel drugs during their storage. As SLNs sit in storage, the formation of a regularly structured crystal lipid core causes the expulsion of drug molecules from the core. A new class of lipid NPs has been developed to overcome this; the nanostructured lipid carriers (NLCs) consist of an irregular solid-lipid crystal matrix that contains a liquid lipid (oil) core in which drugs can be [62]. Although there are no clinical trials employing organic NPs to deliver genes to treat inherited retinal diseases yet, Sun et al. showed that nanolipids integrated into DNA can efficiently deliver the *Rpe65* gene into the retinal pigmented epithelium in the Leber congenital amaurosis (LCA) model of *Rpe65*^−/−^ mice to restore vision [63]. DNA nanoparticles administrated via suprachoroidal is comparable with that injected into subretinal space, resulting in comparable transfection of retina and RPE-choroid in rabbits [64] The irregularity of the crystal lipid core facilitates drug storage and substantially decreases drug expulsion during storage time. Because of this, NLCs generally have higher loading capacities compared to SLNs and have a lower frequency of drug expulsion during storage [61]. 

Adding to the effectiveness of gene transduction, the protamine-DNA core can contain various peptides, such as nuclear localization signals (NLS), transactivator of transcription (TAT), and other functionalizing units. These elements ensure the DNA is localized to the nucleus and that transcription of the carried gene occurs. These additions remarkably increase the transduction efficiency of the particles. For example, the addition of target peptides serves to bind membrane-bound receptors of target cells and facilitate endocytosis. In addition to their use in ocular gene therapy delivery and delivery of conventional ophthalmic drugs, these particles have recently gained acclaim as the vectors used to deliver viral mRNA for the spike protein of the SARS-CoV-2 virus in the Moderna and Pfizer-BioNTech vaccines.

#### 2.2.2. Polymer Nanoparticles and Peptide Nanoparticles (PNPs) 

Polymer nanoparticles and peptide nanoparticles (PNPs) are being investigated as an alternative to lipid-based NPs for the compaction and delivery of DNA (Figure 3). Of many peptides, proteins, and polymers, the most ideally suited to this task is poly-lysine (K). The positive charge of K allows it to stabilize anionic nucleic acids. The ability of poly-lysine NPs to condense DNA and efficiently transduce cells was demonstrated as early as 1996 [65]. Other polymers have been developed for use in PNPs, such as poly-lactic acid (PLA), poly-cryanoacrylate (PCA), and poly-D-D-lactide-co glycolide (PLGA) [66,67,68]. The biodegradable PLGA nanoparticles conjugated with peptide (Gly-Arg-Gly-Asp-Ser-Pro-Lys)-carrying transgene-encoding anti-VEGF intraceptors efficiently decrease the neovascularization and lymphangiogenesis in mice cornea [69]. Recently, much attention has been given to synthetic polyamines, such as polyethylenimine (PEI), as well as polysaccharides, including inulin and chitosan. PEI has been widely explored by numerous investigators as a gene-delivery vector [70]. Biodegradable PNPs are advantageous in that there is no concern of accumulation of NP material following drug delivery [71]. There are countless possible formulations of PNPs, all of which confer different properties to the particle and affect transduction efficacy, cell-type specificity, and potential toxicity.

Currently, the most popular PNP for ocular drug delivery are CK_30_PEG_10k_ NPs, which are created by PEGylation of polyethylene glycol (PEG) to poly-lysine. PEGylation is a common modification used to increase the half-life, reduce the immunogenicity, increase the transduction efficiency, and improve the biological activity of drug delivery system (DDS) [72]. PEGylation also improves the stability of compacted DNA within the NP and prevents significant aggregation of the particles during storage. CK_30_PEG NPs have been used to deliver the *Rds* gene into the retina of a mouse model of retinitis pigmentosa [73]. In 2012, CK_30_PEG was used to deliver the coding sequence for the ABCA4 gene to *Abca4^−/−^* mice and stable expression was noted after 8 months of injection [74]. 

#### 2.2.3. Dendrimers 

Dendrimers are organic polymers that consist of a central core from which ‘branches’ extend outwards to form a spherical shape (Figure 3). These particles are advantageous for ocular drug delivery due to their high loading capacity, versatility of outer functional motif, and controlled drug release. Like liposomes, dendrimers can be made water-soluble by the addition of hydrophilic groups to the outer branches that then interact with water. The ends of these polymeric branches can contain a variety of functional groups to which hydrophilic drugs can be conjugated. Dendrimers using poly(amidoamine) (PAMAM) as their structural motif have become commonly used for ocular gene therapy as they are commercially available. Dendrimers are unique in their ability to buffer endosomal acidification. In addition to their high buffering capacity, they are able to promote an osmotically driven rupture of endosomes, facilitating gene transfer efficiency. 

Currently, there is a phase 1 clinical trial evaluating the safety, tolerability, and pharmacokinetics of D-4517.2, a hydroxyl dendrimer that delivers a VEGFR tyrosine kinase inhibitor to healthy volunteers via subcutaneous administration (NCT05105607). This, along with other emerging drug delivery systems, is shown in Table 4.

#### 2.2.4. Nanoemulsions

Nanoemulsions are a special type of emulsion, which is a mixture of two, normally immiscible liquids, such as oil and water, with one or more surfactants. Nanoemulsions are kinetically stable but thermodynamically unstable. The thermodynamic instability allows for the separation for emulsion over time. The main advantages of using nanoemulsions as gene therapy vectors include enhanced bioavailability and absorption due to the high surface area imparted by the small droplet size. Since the late 1990s, work has been done to optimize cationic nanoemulsions (CNEs) and improve the transfection efficiency and stability of nucleic acids in the CNE. Factors that affect the biologic viability of CNEs include cytotoxicity (mainly a function of the cationic lipids used), resistance to endonuclease-mediated degradation, in vivo particle distribution, and gene transfection efficiency [75]. Although they are relatively non-toxic, the nature of the cationic surfactants can cause some toxicity due to their ability to disrupt cell membranes [76]. This can still be a marked improvement over other common preservatives, such as benzalkonium chloride (BAK). A cationic nanoemulsion containing latanoprost (Catioprost^®^) was shown to have equivalent efficacy and a superior safety profile to Xalatan^®^, an aqueous BAK-preserved latanoprost solution [77].

#### 2.2.5. Lipid-Based Nanoparticles

Although SLNs have emerged as the dominant lipid-based carrier system, nanomicelles; liposomes; and niosomes have also been used in ocular drug delivery. They are structurally similar and all feature spherical configurations of lipids with hydrophilic and hydrophobic components. Ideally, the materials utilized to prepare lipid-based NPs should be biodegradable and/or biocompatible. Biodegradation eliminates the inactive polymers from ocular tissue. In addition to concern about biodegradation, it is essential that the degradation products are not toxic or inflammatory to ocular tissue as some lipid-based carriers have been shown to trigger a strong inflammatory process [78].

Proper delivery must also account for the kinetic and thermodynamical stability of NP, in addition to precorneal retention time, to ensure that there is a limited loss of formulation via precorneal clearance mechanisms. In addition to stability within the precorneal environment, reflux tearing and vitreous humor environment must be considered for proper delivery [79]. Reflux tearing can remove a significant amount of drug; this issue can be avoided by maintaining the appropriate size, isotonicity, and osmolarity of the solution [80]. Additionally, vitreous humor is relatively stagnant, and the release rate depends on the physical stability of the NP.

#### 2.2.6. Nanomicelles

Nanomicelles are colloidal drug delivery systems that assemble instantaneously in solutions. Nanomicelles are ideal for the delivery of hydrophobic drugs to the eye by prolonging drug retention times due to their hydrophilic surface. Nanomicelles can be broadly classified as surfactant nanomicelles and polymeric nanomicelles. Polymeric micelles are typically characterized as more stable, whereas surfactant nanomicelles aggregates are weak and susceptible to physical instability upon dilution [79]. The potential for nanomicelles for sustained drug release longer than a few days is limited, although this still represents a large improvement over conventional eye drops.

Surfactant nanomicellar formulations have been used for more ideal diffusion of topically delivered drugs through the cornea resulting in improved bioavailability. In 2019, positively charged nanomicelles were used to deliver tacrolimus, a hydrophobic macrolide immunosuppressant. The research hypothesized that due to the presence of the negatively charged mucin layer on the outer surface of the eye, the addition of positively charged peptides to micelles should prolong the interface of the solution with the eye. The authors found that this formulation was able to significantly improve tacrolimus retention on the eye surface and improve corneal permeability and bioavailability both ex vivo and in vivo. Furthermore, the total immunosuppressant effect on the eye was observed to be larger than conventionally delivered tacrolimus [81].

#### 2.2.7. Liposomes 

Liposomes are self-assembling spherical vesicles composed of cationic lipids, which can encapsulate hydrophilic particles, such as DNA, small molecules, and biologics, in an aqueous core (Figure 4). Liposomes enter the cell by either phagocytosis or endocytosis. In the clathrin-mediated endocytic pathway, endosome components are usually degraded as the endosome is transformed into a lysosome. However, the use of pH-dependent cationic lipids can facilitate escape from endosomes following endocytosis by selectively becoming cationic at a low pH (such as inside lysosomes). Recently, the use of triggers, such as light, pH, heat, and ultrasound waves, have been used to disrupt the lipid bilayer of the vesicles to improve their release efficiency. 

Liposome-mediated subconjunctival injection of the *BAI1-ECR* gene has shown to effectively reduce experimental corneal neovascularization corneal angiogenesis in rabbits by Yoon et al. in 2005 [82]. Although there has not been much progression in loading gene therapy products in recent decades, liposome has been advanced for pharmaceutical drug delivery. In 2020, Dos Santos et al. explored the role of besifloxacin integrated into liposomes to explore the influence of a cationic liposome delivery system for bacterial conjunctivitis. The liposome integration formulation showed higher permeation than the control (Besivance). The study concluded that besifloxacin incorporation into positively charged liposomes improved passive topical delivery and can be an effective strategy to improve topical ophthalmic treatments [83]. 

There are several clinical trials evaluating the safety and efficacy of liposomal-delivered drugs to the eye, including two trials evaluating the performance of subconjunctival injections of liposomal latanoprost for the treatment of primary open-angle glaucoma (NCT01987323, NCT02466399). The largest clinical trial involving liposomes to date is a Phase 4 trial, which enrolled 200 participants and evaluated the safety and efficacy of liposomal ozone-based solution (OZODROP^®^) in preventing ocular infections following cataract surgery (NCT04087733).

#### 2.2.8. Niosomes 

Niosomes are similar to liposomes in that they are also composed of a lipid bilayer. In niosomes, the bilayer is composed of non-ionic, single-tailed amphiphilic surfactants that surround an aqueous core (Figure 4). This allows for the delivery of both hydrophilic drugs in the aqueous core and lipophilic drugs embedded in the vesicular bilayer [61]. Advantages of niosomes over liposomes include improved chemical stability, storage time, and sustained drug delivery [84]. Furthermore, niosomes are biodegradable and non-immunogenic, and functionalization of the surface hydrophilic head groups is possible [85].

Gene delivery to the retina using niosomes has been investigated by Puras et al. Using Tween 80 as the surfactant, 2,3-di(tetradecycloxy)propan-1-amine as a cationic lipid, and squalene as a helper lipid, the group was able to deliver a pCMS-EGFP plasmid (prepared with the O/W emulsification method) to rat retinas in vivo [86]. More recently, Qin et al. used hyaluronic acid-modified cationic niosomes composed of Tween 80/squalene/1, 2-dioleoyl-3-trimethylammonium-propane (DOTAP) to deliver eGFP to RPE cells with a 6–6.5 times higher than eGFP alone [87].

## 3. Carrier Technology for Ocular Therapeutics

As advances have been made in ocular vectors for gene therapy, novel and non-novel carrier systems have been modified to better maintain therapeutic drug concentration at target sites, overcome various ocular barriers, reduce drug frequency, and enhance drug bioavailability. Human ocular developed, in part, to prevent foreign and toxic substances from reaching the tissues of the eye. Meanwhile, these barriers serve as challenges for drug administration as they often reduce ocular absorption and, therefore, bioavailability. Additionally, factors, such as tear production and blinking, reduce the absorption of many formulations [88,89,90]. In the anterior segment of the eye, chemical and mechanical barriers produced by the cornea prevent the passage of any foreign bodies, including drugs, into the eye [91]. The scleral, choroidal, and retina epithelial in the posterior segment play a major role in limiting bioavailability in the posterior of the eye [92]. Despite these barriers, there are many ocular drug delivery systems that continue to be developed and modified to further increase bioavailability and tissue targeting.

### 3.1. Conventional Carrier Systems 

#### 3.1.1. Topical Ophthalmic Solutions

Topical ophthalmic solutions (eye drops) have been the mainstay of treatment and prevention of a variety of ocular pathologies, including glaucoma, microbial infection, inflammatory conditions, and dry eye syndrome for many years. Topical drops serve as a safe, noninvasive, and simple means of delivering medications to the anterior segment of the eye with low systemic absorption [93]. 

While topical drug administration remains the most common route, it does face certain challenges, compared to other methods. Topical drugs typically have very low bioavailability, with less than 5% of administered drug reaching target tissues [94]. This is the result of tear formation, rapid absorption from the conjunctival vasculature, and poor patient compliance [95].

Current research has focused on improving the efficacy of topical administered drugs with anterior segment tissue targets. Recently, Vicente-Pascual et al. have developed a means of topically administering SLNs, combined with ligands, which included protamine, dextran, and hyaluronic acid, to increase IL-10 production in the setting of corneal inflammation [96]. 

#### 3.1.2. Cyclodextrins

Ophthalmic solutions have been improved by the formation of cyclodextrins (CDs) and the addition of permeation enhancers. CDs allow hydrophobic drugs to complex around the structure and result in increased solubility and bioavailability in the ocular environment. These complexes also reduce tissue inflammation while increasing corneal residence time [97]. CDs may provide a unique solution to overcome the natural barriers of the eye by increasing solubility and permeability, allowing topically administered drugs to penetrate to the posterior segment [98]. Most recently, αCDPPRs containing natamycin were investigated using single or mixed micelles of Pluronic P_103_ and Soluplus [79]. Key to the future development of CDPPRs was the finding that CDPPRs made from mixed micelles of both polymers showed intermediate drug permeability, suggesting that combining copolymers may be a way to fine-tune the drug release and permeability profiles of CDPPR-based formulations [99].

#### 3.1.3. Suspensions

Eye drops of poorly soluble drugs are frequently formulated as suspensions. Suspensions finely disperse insoluble drug particles in an aqueous solution of solubilizing agents. Suspensions are beneficial, as the precorneal cavity retains drug particles in suspension, thereby enhancing the contact time of the drug. The bioavailability of drugs in suspension depends on both the retention and the dissolution of drug particles in tears [100]. Additionally, a recent study showed that both viscosity and size have a clear impact on ocular absorption [101]. Toropainen et al. compared the FDA-approved indomethacin suspension (Indom^®^ 0.5%) to experimental suspension (INDO1, INDO2, INDO3, INDO4, INDO5, INDO6), which has a different size and viscosity. In this study, median particle size (d50) categories were 0.37–1.33 and 3.12–3.50 µm and particle viscosity levels were 1.3, 7.0, and 15 mPa·s. The results concluded that higher viscosity increased ocular absorption 3.4–4.3-fold for suspensions with similar particle sizes. Additionally, the bioavailability range for the suspensions was about 8-fold, with small particles yielding a higher concentration of dissolved indomethacin in the tear fluid, thereby leading to improved ocular bioavailability [102]. 

### 3.2. Innovative Carrier Technology

#### 3.2.1. Punctal Plugs

Punctal plugs are biocompatible instruments inserted in the tear duct opening (punctum). Punctal Plugs are often used to block tear drainage and can also be used to provide controlled drug release for up to 180 days to the anterior segment of the eye [103]. Recent trials by Ocular Therapeutix have studied punctal plugs efficacy in delivering a variety of drugs. In 2018, Ocular Therapeutix successfully completed Phase III trials and received FDA approval of Dextenza (dexamethasone punctal plug insert) for the treatment of post-surgical ocular inflammation and pain. However, in 2019, a Phase III trial of OTX-TP (travoprost punctal plug insert) for the treatment of glaucoma failed to show statistically significant superiority of mean reduction in IOP, compared with the placebo [104].

#### 3.2.2. Drug-Eluting Contact Lenses

Contact lenses were first evaluated for their use as drug delivery systems in 1965 by J. Sedlavek [105]. The embedding of drugs in contact lenses for sustained, continuous-release or pulsatile release in response to some stimuli offers some unique advantages among drug delivery systems [106]. Compared with other drug delivery systems, contact lenses have a high rate of patient compliance due to their superb biocompatibility. Polymeric hydrogels, surface-modified polymeric hydrogels, and molecularly imprinted polymeric hydrogels have all been used to absorb and subsequently slowly release drugs while worn. A list of drug-soaked and imprinted contact lenses with ocular drugs has been covered in a comprehensive review by Xu et al. [106].

## 4. Advancements in Intraocular Administration Routes and Systemic Drug Delivery

Static and dynamic barriers of the eye render the drug penetration to both the anterior and posterior segment a major challenge via systemic delivery. With rapid advancements in ocular pharmacology, there has been a growing body of studies to improve drug bioavailability, safety, stability, and tolerability of drug administrated, bypassing the barriers. Despite the fact that there is less progression regarding the systemic administration targeting eye diseases specifically, the new developed treatment regimen to adopt systemic routes targeting central serous chorioretinopathy and the secondary eye disorders following systemic diseases are reviewed.

### 4.1. Anterior Segment

#### 4.1.1. Subconjunctival Administration

Subconjunctival drug administration has been used to decrease the injection risks and treatment burden seen in current intravitreal therapies. This route is less invasive and can provide sustained drug delivery to the anterior segment of the eye for longer durations than intravitreal drug administration (Figure 5) [96,107]. Subconjunctival injection allows for the bypass of structures that impede drug permeability, such as the cornea and conjunctiva, allowing for improved penetration [108]. Although those barriers are no longer an issue with subconjunctival injection, the systemic absorption through blood and lymphatics remains, which is particularly an issue for the administration of free drug [109]. 

In 2021, Zhang et al. evaluated the therapeutic potential of subconjunctival tumor necrosis factor-alpha-treated bone marrow-derived mesenchymal stem cells in rat corneal alkali burns. This intervention increased corneal epithelium repair, as well as decreased inflammatory cell infiltration and fibrosis, compared to control. This study suggested that it may be mediated through the upregulation of prostaglandin-endoperoxide synthase 2 and TNF-inducible gene 6 protein, which may serve as potential treatment modality [110].

A recent case report about the management of post-cataract surgery cystoid macular edema (CME) discussed the use of subconjunctival interferon alpha 2b for refractory disease [111]. Despite clinically significant CME being reported in only 0.1–2.35% of cataract surgeries, it remains the leading cause of decreased vision after cataract surgery. Although the majority of cases of CME resolve spontaneously, there is still a need for management of chronic CME after cataract surgery [112]. This case report details the first use and potential of subconjunctival interferon alpha 2b as alternative treatment for refractory CME. 

#### 4.1.2. Intrastromal Administration

In corneal diseases that involve the deeper stroma, topical and subconjunctival drug administration show decreased effectiveness, which is particularly true for infectious corneal diseases, such as fungal keratitis. Moreover, there are few antifungal drugs that can adequately penetrate the deeper layers of the cornea. To increase drug penetration, intrastromal injections, which can effectively deliver drugs to the site of pathology, have been utilized clinically for the past decade. The use of intrastromal amphotericin B, voriconazole, and natamycin have been reported as a safe, effective adjuvant therapy in the management of recalcitrant fungal keratitis [113,114]. 

The intrastromal administration of AAVs has also recently been used in a canine model to treat mucopolysaccharidosis type I, a lysosomal storage disorder in which patients can experience corneal clouding. Researchers used AAV8G9-expressing IUDA, an enzyme required for glycosaminoglycan degradation, to reverse corneal opacity within one week of intrastromal administration, which persisted for at least 25 weeks [48]. Further evaluation of intrastromal AAV-containing IUDA showed minimal toxicity and inflammation in rabbits [115].

#### 4.1.3. Intracameral Administration

Intracameral injections involved the administration of therapeutic agents to the anterior segment of the eye though the cornea (Figure 5) [116]. This technique is often used to administer antibiotics to the anterior segment during cataract surgery, as well as to introduce IOP-lowering agents to the area. Currently, there is one FDA-approved intracameral implant, Bimatoprost sustained release, which allows for the maintenance of drug concentration for up to four months after insertion [117]. 

### 4.2. Posterior Segment

#### 4.2.1. Intravitreal Administration

The delivery of therapeutic agents to the eye by means of intravitreal injection has been widely used in clinics to treat ocular diseases. From 1997 to 2001, there were less than 5000 intravitreal injections performed each year worldwide [118]. As of 2016, there were 5.9 million intravitreal injections performed in the United States alone [119]. This increase is the result of the introduction of humanized monoclonal antibodies against vascular endothelial growth factor (VEGF), which has become first-line therapy in the treatment of neovascular age-related macular degeneration, diabetic macular edema, and other vascular pathologies of the eye. Despite their influence on the treatment of ocular diseases, intravitreal injections pose the risk of endophthalmitis, retinal detachments, cataract formation, and ocular hypertension, among others [120]. Furthermore, the treatment burden associated with injection of monoclonal anti-VEGF agents can be very high, as many patients require treatment every four weeks. 

#### 4.2.2. Subretinal Administration

Subretinal drug administration is being investigated for gene therapy, particularly regarding the treatment of inherited retinal diseases. Although this technique directly exposes the retina to therapeutic agents, it is arguably the most invasive administration route as it requires vitrectomy prior to administration in an operating room, which put the patient at risk of complications (Figure 6) [121]. 

As discoveries are made in subretinal gene therapy delivery, there is an increasing need for the development of new techniques that safely achieve required drug concentrations in the posterior segment of the eye. Recently, there has been interest in using robotic assist devices for subretinal injections. A recent clinical trial compared the use of robot assist devices against vitreoretinal surgeons not using assistive devices (NCT03052881). The results showed that both groups were able to successfully complete the procedure, with the results and surgery times similar between both groups. Although the difference in results for each arm of the study was not statistically significant, this comparison confirmed that assistive devices may have utility in this procedure and could be utilized to overcome the limit of human capabilities in surgery [122]. 

#### 4.2.3. Suprachoroidal Administration

Suprachoroidal (SC) space injection technique involves the delivery of medication to a 30-micron thick area of tissue between the sclera and choroid using hollow microneedles (Figure 6). Importantly, SC injections are able to be performed in the clinic, in contrast to subretinal injections. This technique would allow for a more direct means of delivering medications to the posterior segment of the eye in the clinic, minimizing drug loss in the process [123,124]. This technique is not currently being employed in clinical practice widely; however, current clinical trials are exploring the use of suprachoroidal triamcinolone in the treatment of serous retinal detachments, macular edema secondary to vascular occlusions and DME, and non-infectious uveitis.

Notably, Regenxbio is utilizing suprachoroidal injections to deliver its gene therapy RGX-314, an AAV8 vector containing a transgene coding for an anti-VEGF fab, for the treatment of nAMD and diabetic retinopathy. There are concurrent clinical trials evaluating the safety and efficacy of RGX-314 when administered using IVT or subretinal injections as well. 

#### 4.2.4. Systemic Delivery 

In addition to the numerous increased toxicities that can result from the systemic delivery of viral and non-viral vectors, there are robust anatomic barriers that prevents substances in systemic circulation from reaching the posterior segment of the eye. The blood–retinal barrier (BRB) is a part of the blood–ocular barrier (BOB) and functions analogously to the blood–brain barrier (BBB) preventing large, hydrophilic molecules from entering the eye from the blood. These barriers result in the inability of many drugs to be delivered in a way that provides therapeutic concentrations at the target site without reaching toxic levels systemically. For this reason, many drugs that can be delivered orally have been adapted to be delivered topically. Examples of this include antihistamines for allergic conjunctivitis and antiviral drugs in the treatment of viral infections, such as varicella zoster and herpes simplex. 

Compared to other routes of administration, there is less investigation to improve orally administered drugs. Recently there have been studies exploring orally administered drugs for a noninvasive means of medically managing central serous chorioretinopathy (CSCR). Of the many drugs investigated, the mineralocorticoid receptor antagonist eplerenone has shown the most potential [125,126]. Since being implicated in the pathogenesis of CSCR, mineralocorticoid receptor blockade has been studied as a means of managing CSCR, but the value of eplerenone in clinical practice is highly disputed [127,128,129]. 

Recently there has been investigation into the use of subcutaneously administered medication for the treatment of ocular disease (Figure 6). For instance, in recent years subcutaneous tocilizumab has been used instead of an intravenous administration in the treatment thyroid eye disease, giant cell arteritis, and neuromyelitis optica [130,131]. Although intravenous tocilizumab infusion is generally well tolerated, it is very time consuming for patients and an expensive off-label treatment. Furthermore, in 2021, tocilizumab was granted an emergency use authorization for the treatment of individuals hospitalized due to severe COVID-19 infection. As a result, there was interest to reduce the burden on patients and help to avoid potentially limited supplies of the medication. A recent study investigated the use of subcutaneous tocilizumab, compared to prednisolone taper, showed that weekly and biweekly subcutaneous tocilizumab injections achieved remission at 52 weeks in 56% and 53% of patients. Comparatively, 14% of patients that underwent a 26 week taper and 18% of patients that underwent a 52 week taper were in remission at 52 weeks [132]. As mentioned in the subsection on dendrimers, there is also a phase 1 clinical trial investigating the safety, tolerability, and pharmacokinetics of D-4517.2, which is a hydroxyl dendrimer that delivers VEGF tyrosine kinase inhibitor by subcutaneous injection (NCT05105607). 

## 5. Conclusions

Effective ocular drug delivery has posed a challenge to ophthalmologists for decades. Despite the obstacles, the compartmentalization, accessibility, and immune privilege of the eye still make it an excellent candidate for gene therapy research. The increase in the development and availability of gene therapy products has prompted the study of viral and non-viral-based vectors administration and expanded the scope of drug delivery. Transgene construct optimization, vector selection, and vector engineering are now essential to further development and eventual approval of disease-specialized gene therapy products. Although both viral and non-viral vectors have advantages and disadvantages, current research trends point to adeno-associated viruses (AAV) as an area of great interest in ocular drug administration. AAVs are efficient at transducing photoreceptors (PRs) and retinal pigment epithelium (RPE) cells, and future studies should seek to increase capacity while reducing immunogenicity [133,134]. 

With advances in ocular vector technology, carrier system technology and ocular therapeutic administration routes will improve to maintain therapeutic concentration at the target site, reduce administration frequency, overcome various ocular barriers, and enhance drug bioavailability. The unique anatomy and natural barriers of the eye present challenges to such systems, but advancements in novel and non-novel delivery systems continue to show promising results in multiple clinical trials. Topical solutions have increased their effectiveness through the development of cyclodextrins, the addition of permeation enhancers, and delivery via suspensions or emulsions [76,98,102,135]. Although topical instillation is the least invasive treatment option, administration via topical routes is not effective against diseases that affect posterior tissues, such as age-related macular degeneration, diabetic retinopathy, posterior uveitis, and retinitis due to glaucoma. 

Furthermore, topical drugs typically have low bioavailability and are often used incorrectly by patients [30]. Intravitreal or subretinal injections are currently used as delivery systems for the posterior of the eye but with risk. Suprachoroidal space injection is emerging as a promising new administration route to minimize drug loss [107]. Novel lipid-based solutions that encapsulate the drug are promising injectable solutions because of their high loading capacity, versatility of outer functional groups and controlled drug release, but must account for kinetic and thermodynamic stability, precorneal retention time, and toxicity to ensure proper delivery [61,96,136]. Subconjunctival administration has the potential to decrease risks of injection reaching the posterior segment of the eye, and while there are two polymer implants available on the market in the United States, more research is needed to robustly assess clinical viability.

Despite the challenges of ocular drug delivery, advancements in gene therapy for ocular diseases are occurring at a rapid pace. The combined efforts of academia, industry, and regulatory agencies are continually improving the state-of-the-art and safer, more efficacious drug delivery systems are being developed, which will continue to improve therapeutic outcomes.

## Figures and Tables

**Figure 1 pharmaceutics-14-01931-f001:**
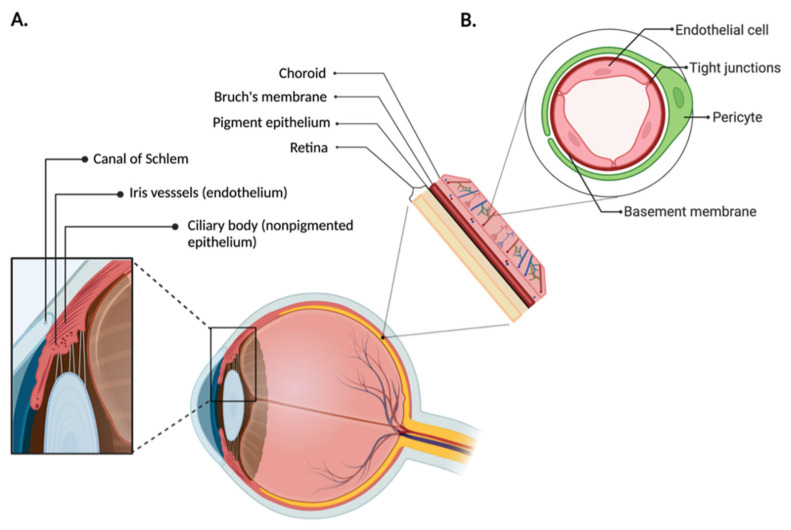
Anatomical barriers of the eye. (**A**) In the anterior of the eye, the blood–aqueous barrier, consisting of the iris/ciliary blood vessels and nonpigmented ciliary epithelium, limits access to the anterior of the eye and prevents therapeutic entry to the intraocular environment. (**B**) In the posterior of the eye, the blood-retinal barrier, comprised of the retinal capillary endothelial cells and retinal pigment epithelium cells, prevents therapeutics from entering the posterior segment from the bloodstream.

**Figure 2 pharmaceutics-14-01931-f002:**
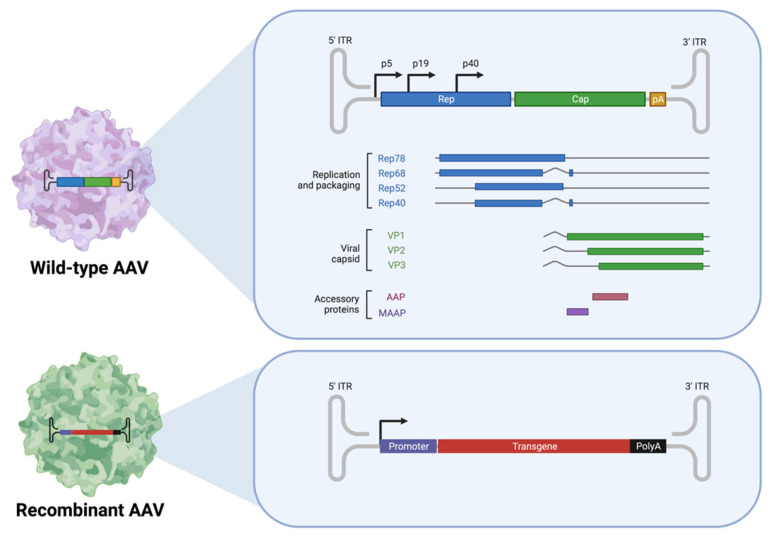
Schematic of the wild type and recombinant AAV genome. The wild type AAV genome comprised four known open reading frames, rep (blue), cap (green), MAAP (burgundy), and AAP (purple), flanked by inverted terminal repeats (ITRs, grey color). The rep gene encodes four regulatory proteins: Rep78, Rep68, Rep52, and Rep40. Cap gene encodes viral proteins, VP1, VP2, and VP3. Rep and cap genes are removed from the genome of recombinant AAV, instead the transgene expression cassette was inserted, flanked by ITRs. Created with BioRender.com.

**Figure 3 pharmaceutics-14-01931-f003:**
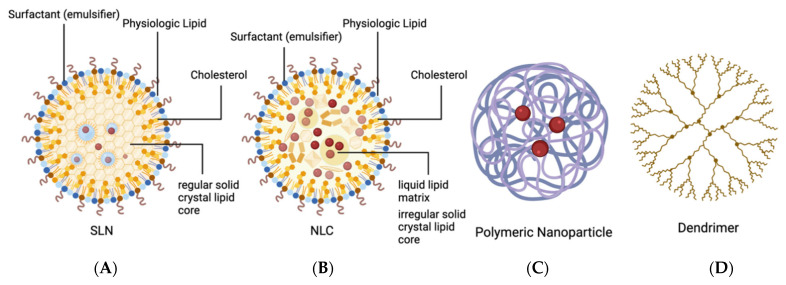
The structure of organic nanoparticles (NPs). (**A**) Solid lipid nanoparticles (SLNs) range from 50 to 1000 nm and consist of a solid crystal lipid core that is stabilized by surfactant, which acts as an emulsifier. Rather than a simple phospholipid bilayer, the exterior of an SLN feature a bilayer of physiologic lipids, such as triglycerides and cholesterol. (**B**) Nanostructured lipid carriers (NLCs) range from 30 to 100 nm in size and have a similar outer layer to SLNs and contain a surfactant that acts as an emulsifier, but they utilize an irregular solid lipid crystal matrix and a liquid lipid (oil) core. (**C**) Polymeric nanoparticles range from 1 to 1000 nm. They utilize a polymetric core that can be loaded with active compounds to produce countless variations of the nanoparticle. (**D**) Dendrimers are organic polymers, which consist of a central core from which ‘branches’ extend outwards to form a spherical shape. They range from 1 to 10 nm in size.

**Figure 4 pharmaceutics-14-01931-f004:**
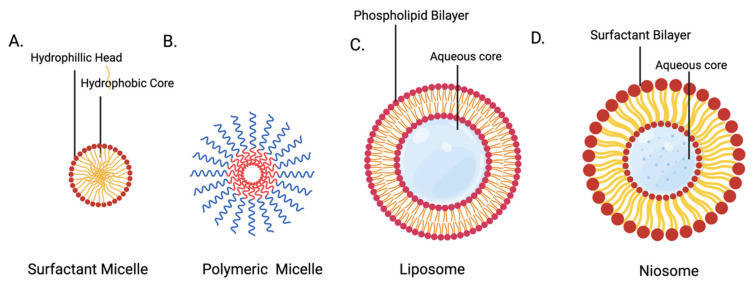
The structures of nanomicelles., liposomes, and niosomes (**A**) Surfactant micelles are grouped molecules of amphipathic lipids that create a hydrophobic core. Their sizes can vary but generally range from 10–100 nm. (**B**) Polymeric micelles contained amphiphilic copolymer that, when exposed to water, automatically forms a core/shell structure that can be loaded with insoluble drugs. They are typically between 10–100 nm in size. (**C**) Liposomes consist of an amphiphilic phospholipid bilayer that encloses hydrophilic substances and are typically between 25 and 2500 nm in size. (**D**) Niosomes are composed of non-ionic surfactant that form a vesicle to transport aqueous material. They are typically between 25 and 100 nm in size.

**Figure 5 pharmaceutics-14-01931-f005:**
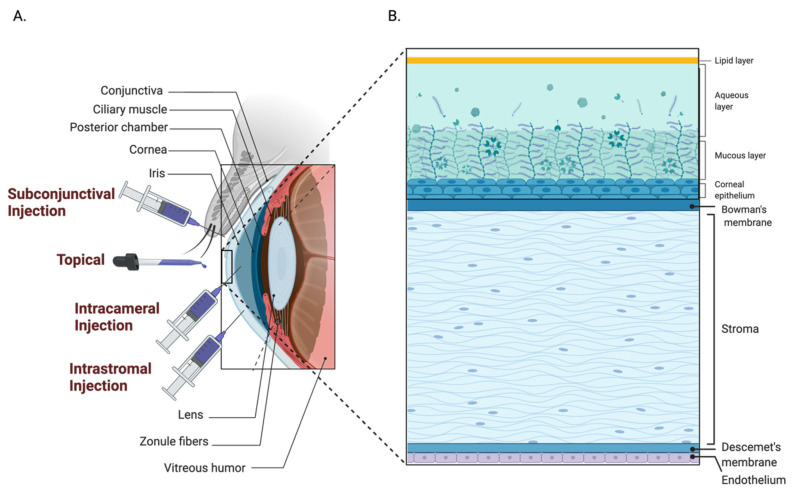
Drug administration routes to the anterior segment of the eye. (**A**) Subconjunctival injections are a type of periocular route of administration where the drug is injected under the conjunctiva (epibulbar) or underneath the conjunctiva lining the eyelid (subpalpebral). Intracameral injection delivers medication directly to the anterior chamber/aqueous humor of the eye. Topical administration is most used in drop form and delivers the solution to the exterior of the eye. Intrastromal injection administers the drug directly to the thick, fibrous stroma of the cornea. (**B**) Figure 3B depicts the layers of the tear film and cornea. The tear film stretches from the lipid layer to the mucous layer and the cornea from the corneal epithelium to the endothelium.

**Figure 6 pharmaceutics-14-01931-f006:**
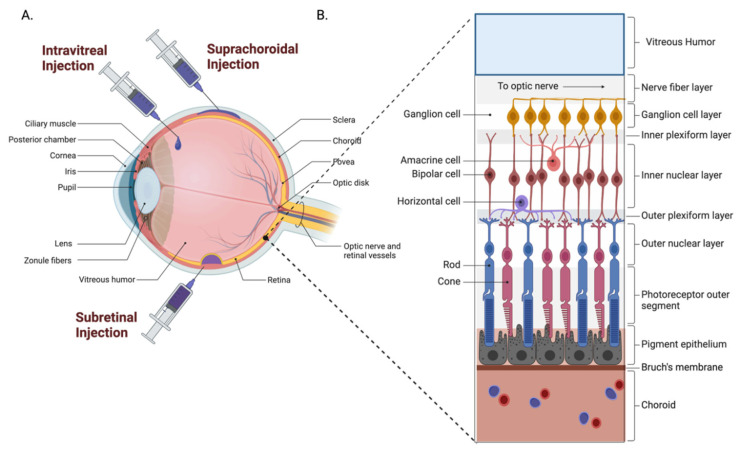
Drug administration routes to the posterior anatomy of the eye. (**A**) Intravitreal injections administer the drug directly to the vitreous humor. Suprachoroidal injections deliver directly into the suprachoroidal space. Implants are non-biodegradable systems often anchored to the sclera or injected into the vitreous that release drugs at a predetermined rate. Subretinal injections target the subretinal space or the area directly between retinal pigment epithelium (RPE) cells and photoreceptors. (**B**) The posterior anatomy of the eye consists of the vitreous humor and the retina, which contains layers of terminally differentiated cells used for light perception.

**Table 1 pharmaceutics-14-01931-t001:** Current and past clinical trials using AAVs as vectors for gene therapy.

Conditions	Sponsor	AAV Serotype	Gene Therapy Product	Transgene	Administration Route	Clinical Trial Status	NCT Number(s)
Neovascular AMD	Regenxbio Inc.	AAV8	RGX-314	mAb fragment, anti-VEGF	Suprachoroidal injection(s)	Phase 1/2a, 2, 2/3, long-term follow-up	NCT03066258, NCT04514653, NCT05210803
	Regenxbio Inc.	AAV8	RGX-314	mAb fragment, anti-VEGF	One-time intravitreal injection	Phase 2/3	NCT04704921
	Regenxbio Inc.	AAV8	RGX-314	mAb fragment, anti-VEGF	One-time subretinal injection	Phase 2, long-term follow-up	NCT04832724, NCT03999801
	Adverum Biotechnologies, Inc.	AAV7	ADVM-022	aflibercept	One-time intravitreal injection	Phase 1, long-term follow-up	NCT03748784, NCT04645212
	Genzyme/Sanofi	AAV2	AAV2-sFLT01	sFLT-1	One-time intravitreal injection	Phase 1	NCT01024998
	Lions Eye Institute	-	rAAV.sFlt-1	sFLT-1	One-time subretinal injection	Phase 1/2	NCT01494805
	Gyroscope Therapeutics Limited	AAV2	GT005	Complement factor I (CFI) gene	One-time subretinal injection	Phase 2	NCT03846193
	4D Molecular Therapeutics	R100 capsid	4D-150	Anti-VEGF-C miRNA and codon-optimized sequence encoding aflibercept	One-time intravitreal injection	Phase 2	NCT05197270
Diabetic macular edema	Adverum Biotechnologies, Inc.	AAV7	ADVM-022	aflibercept	One-time intravitreal injection	Phase 2	NCT04418427
Diabetic retinopathy	Regenxbio Inc.	AAV8	RGX-314	mAb fragment anti-VEGF	One or two suprachoroidal injections	Phase 2, long-term follow-up	NCT04567550, NCT05296447
X-linked retinitis pigmentosa	MeiraGT UK II Ltd.	AAV2/5	AAV2/5-RPGR	RPGR coding sequence	One-time subretinal injection	Phase 1/2, 3	NCT03252847, NCT04671433
	NightstaRx Ltd./Biogen	AAV8	BIIB112	RPGR coding sequence	Six-time subretinal injection	Phase 1/2	NCT03116113
	4D Molecular Therapeutics	R100 capsid	4D-125	Codon-optimized *RPGR* gene	One-time intravitreal injection	Phase 1/2	NCT04517149
	Applied Genetic Technologies Corp.	AAV2tYF	AGTC-501 (rAAV2tYF-GRK1-hRPGRco)	G Protein-Coupled Receptor Kinase 1 (*GRK1*) and *RPGR* coding sequences	One-time subretinal injection	Phase 1/2, 2/3	NCT03316560, NCT04850118
Retinitis pigmentosa	Coave Therapeutics	AAV2/5	AAV2/5-hPDE6B	*PDE6B* gene	Subretinal injection	Phase 1/2	NCT03328130
	STZ eye trial	-	rAAV.hPDE6A	*PDE6A* gene	One-time subretinal injection	Phase 1/2	NCT04611503
	King Khaled Eye Specialist Hospital	AAV2	rAAV2-VMD2-hMERTK	VMD2-hMERTK gene vector	Subretinal injection	Phase 1	NCT01482195
	Nanoscope Therapeutics Inc.	AAV2	vMCO-1	Multi-Characteristic Opsin 1 gene expression. cassette	One-time intravitreal injection	Phase 1/2	NCT04919473
	GenSight Biologics	AAV2	GS030 (rAAV2.7m8-CAG-ChrimsonR-tdTomato)-Medical Device	Channel rhodopsin ChrimsonR-tdTomato gene with Visual Interface Stimulating Glasses	One-time intravitreal injection	Phase 1/2	NCT03326336
	Ocugen	AAV5	OCU400	Nuclear Hormone Receptor (NR2E3) gene	One-time subretinal injection	Phase 1/2	NCT03326336
	Nanoscope Therapeutics Inc.	AAV2	vMCO-101	Multi-characteristic opsin (MCO) gene expression cassette	One-time intravitreal injection	Phase 2	NCT04945772
Choroideremia	University of Oxford	AAV2	rAAV2.REP1	Rab-escort Protein 1 (REP1) coding sequence	Subretinal injection	Phase 1/2	NCT01461213
	Spark Therapeutics	AAV2	AAV2-hCHM (human choroideremia gene, same as REP1)	Rab-escort Protein 1 (REP1) coding sequence	Subretinal injection	Phase 1/2	NCT02341807
	Byron Lam	AAV2	AAV2-REP1	Rab-escort Protein 1 (REP1) coding sequence	Subretinal injection	Phase 2	NCT02553135
	4D Molecular Therapeutics	R100	4D-R100	Codon-optimized Rab-escort Protein 1 (REP1) coding sequence	One-time intravitreal injection	Phase 1	NCT04483440
	STZ eye trial	AAV2	rAAV2.REP1	Rab-escort Protein 1 (REP1) coding sequence	One-time subretinal injection	Phase 2	NCT02671539
	Ian M. MacDonald	AAV2	rAAV2.REP1	Rab-escort Protein 1 (REP1) coding sequence	One-time subretinal injection	Phase 1/2	NCT02077361
Leber congenital amaurosis	Spark Therapeutics	AAV2	LUXTURNA, voretigene neparvovec-rzyl (AAV2-hRPE65v2)	RPE65 gene	One-time subretinal injection	Phase 1, 1/2, 5-year follow-up, 3, 15-year follow-up	NCT00516477, NCT01208389, NCT03597399, NCT00999609, NCT03602820
	MeiraGTx UK II Ltd.	AAV2	AAV2/5-OPTIRPE65	RPE65 gene	One-time subretinal injection	Phase 1/2, long-term follow-up	NCT02781480, NCT02946879
	University College, London	AAV2	tgAAG76 (rAAV 2/2.hRPE65p.hRPE65)	RPE65 gene	One-time subretinal injection	Phase 1/2	NCT00643747
	Applied Genetic Technologies Corp	AAV2	rAAV2-CB-hRPE65	RPE65 gene	One-time subretinal injection	Phase 1/2	NCT00749957
Autosomal recessive Leber congenital amaurosis	Atsena Therapeutics Inc.	AAV5	SAR-439483	GUCY2D gene	One-time subretinal injection	Phase 1/2	NCT03920007
Leber Hereditary Optic Neuropathy	GenSight Biologics	AAV2	GS010 (rAAV2/2-ND4)	ND4 gene (mitochondrial)	One-time intravitreal injection	Phase 3	NCT03293524
	Byron Lam	Self-complementary AAV2	scAAV2-P1ND4v2	ND4 gene (mitochondrial)	One-time intravitreal injection	Phase 1	NCT02161380
	MeiraGTx UK II Ltd.	AAV2/8	AAV2/8-hG1.7p.coCNGA3	CNGA3 gene	One-time subretinal injection	Phase 1/2	NCT03758404
	Applied Genetic Technologies Corp	AAV2	AGTC-402 (rAAV2tYF-PR1.7-hCNGA3)	CNGA3 gene	One-time subretinal injection	Phase 1/2	NCT02935517
	Applied Genetic Technologies Corp	AAV2	AGTC-401 (rAAV2tYF-PR1.7-hCNGB3)	CNGB3 gene	One-time subretinal injection	Phase 1/2	NCT02599922
Variant Late-Infantile Neuronal Ceroid Lipofuscinosis	Amicus Therapeutics	Self-complementary AAV9	scAAV9.CB.CLN6	CLN6 Gene	One-time intrathecal injection	Phase 1/2	NCT02725580
X-linked Juvenile Retinoschisis	National Eye Institute (NEI)	AAV8	AAV8-scRS/IRBPhRS	RS1 gene	One-time intravitreal injection	Phase 1/2	NCT02317887
	Genetic Technologies Corp	AAV2	rAAV2tYF-CB-hRS1	RS1 gene	One-time intravitreal injection	Phase 1/2	NCT02416622

**Table 2 pharmaceutics-14-01931-t002:** Pre-clinical trial gene therapy products for the treatment of glaucoma and corneal dystrophy.

Condition Treated	Gene Therapy Product	AAV Serotype	Delivery Method	Animal Model	Comments/Mechanism	References
Open-angle-glaucoma	scAAV2.CMV.GFP	AAV2	Intracameral injection	NHPs	Resulted in fluorescence in TM for 2 years	[34]
Open-angle glaucoma	AAV2-Shp2 eGFP-shRNA	AAV2	Intravitreal injection	Cav-1 deficient mouse model of glaucoma	Prevented inner retinal injury due to ocular hypertension	[36]
Open-angle glaucoma	AAV2-BMP4	AAV2	Intravitreal injection	Magnetic microbead-induced glaucoma, mouse	Retinal ganglion cell survival was enhanced and the amplitude of the PhNR was restored in ERG	[37]
Open-angle glaucoma	*AAV2(Y444F)-smCBA-hADAMTS10*	AAV2	Intracameral injection	Six *ADAMTS10*-mutant dogs	Treated eyes showed almost complete prevention of extracellular plaque formation	[38]
Open-angle glaucoma	*AAV2-XIAP*	AAV2	Intravitreal injection	Intracameral injections of microbeads, mouse	XIAP overexpression resulted in significant protection of RGCs	[39]
CoNV	AAV8-KH902	AAV8	Intrastromal injection	Alkali burn model, mouse	AAV8 showed superior efficacy to AAV2	[40]
CoNV	scAAV8G9-optHLA-G1 + G5	AAV8	Intrastromal injection	Burn-induced CoNV, rabbit	HLA-G upregulates Treg cells, preventing foreign body rejection	[41]
CoNV	AAV5-decorin	AAV5	Topical	Corneal micropocket assay model of CoNV, rabbit	Decorin is a TGF-β inhibitor	[42]
Corneal Fibrosis	AAV5-Smad7	AAV5	Topical	PRK-induced corneal fibrosis, rabbit	Smad7 is a negative regulator of TGF-β	[43]
Corneal Fibrosis	AAV5-decorin	AAV5	Topical	PRK-induced corneal fibrosis, rabbit	Decorin is a TGF-β inhibitor	[44]
Corneal Transplant Rejection	AAV8-KH902	AAV8	Intrastromal injection	Cornel suture model, rat	CoNV and corneal opacity were decreased, graft survival rate was increased	[45]
HSV-mediated keratitis	scAAV2-LAT	AAV2	Abrasion followed by topical administration	HSV-infected rabbits	Viral reactivation was blocked in 60% or rabbits	[46]
Mucopolysaccharidosis VI	AAV8-ArsB	AAV8	Intrastromal injection and sequential (opposite eye) intrastromal injection	One ArsB homozygous and one heterozygous ArsB feline mutants	Corneal opacity was reversed, and no signs of an inhibitory capsid antibody response observed in opposite eye	[47]
Mucopolysaccharidosis I	AAV8G9-opt-IDUA (AAV8 and 9 chimeric capsid-optimized-*IDUA*)	AAV8	Intrastromal injection	MPS I canine model, homozygous for the *IDUA* gene mutation	Treatment was able to prevent and reverse visual impairment	[48]

**Table 3 pharmaceutics-14-01931-t003:** Corneal dystrophies and their identified gene mutations.

Corneal Dystrophy Type	Gene/Chromosomal Locus	Gene Size	Open-Reading Frame Size	Inheritance Pattern
Avellino Type	*TGFBI*	34.8 kb	2.05 kb	AD
Congenital Endothelial 1	20p11.2–q11.2 locus	unknown	unknown	AD
Congenital Stromal	*DCN*	42.3 kb	1.08 kb	AD
Epithelial Basement Membrane	*TGFBI*	34.8 kb	2.05 kb	AD
Fleck	*PIKFYVE*	92.7 kb	6.29 kb	AD
Fuchs Endothelial, Early Onset	*COL8A2*	29.9 kb	2.11 kb	AD
Fuchs Endothelial, Late Onset	*ZEB1*	211.4 kb	3.37 kb	AD
Fuchs Endothelial, Late Onset 2	*TCF4*	442.6 kb	2 kb	AD
Granular	*TGFBI*	34.8 kb	2.05 kb	AD
Lattice Type I	*TGFBI*	34.8 kb	2.05 kb	AD
Lattice Type II	*GSN*	131.4 kb	2.35 kb	AD
Meesmann	*KRT12*, *KRT3*	5.92 kb, 6.43 kb	1.48 kb, 1.88 kb	AD
Posterior Amorphous	12q21.33 deletion	unknown	unknown	AD
Posterior Polymorphous 1	*OVOL2*	102.2 kb	0.83 kb	AD
Posterior Polymorphous 2	*COL8A2*	29.9 kb	2.1 kb	AD
Posterior Polymorphous 3	*ZEB1*	211.4 kb	3.37 kb	AD
Posterior Polymorphous 4	*GRHL2*	188.7 kb	1.86 kb	AD
Recurrent Epithelial Erosions	unknown	unknown	unknown	AD
Reis–Bücklers	*TGFBI*	34.8 kb	2.05 kb	AD
Schnyder	*UBIAD1*	26.4 kb	1.01 kb	AD
Stocker–Holt	*KRT12*	5.92 kb	1.48 kb	AD
Subepithelial Mucinous	unknown	unknown	unknown	AD
Thiel–Behnke	*TGFBI*	34.8 kb	2.05 kb	AD
Band-Shaped	unknown	unknown	unknown	Unknown
Congenital Endothelial 2	*SLC4A11*	12.1 kb	2.63 kb	AR
Gelatinous Drop-like	*TACSTD2*	1.82 kb	0.97 kb	AR
Macular	*CHST6*	23.4 kb	1.19 kb	AR
Lisch Epithelial	unknown	unknown	unknown	X-linked, dominant
Endothelial X-Linked	Xq25 locus	unknown	unknown	X-linked, unclear

**Table 4 pharmaceutics-14-01931-t004:** Current and past clinical trials using emerging drug delivery system.

Nanoparticle	Drug Carried	Condition Treated	Delivery Method	Sponsor	Clinical Trial Status	NCT Number(s)
Cyclodextrin NP	Dexamethasone	Diabetic macular edema	Topical	King Saud University	Phase 2/3	NCT01523314
D-4517.2 (Hydroxyl Dendrimer)	VEGFR Tyrosine Kinase Inhibitor	AMD	Subcutaneous injection	Ashvattha Therapeutics, Inc.	Phase 1	NCT05105607
TLC399 (ProDex) Multi-layered lipid NP	Dexamethasone	Retinal vein occlusion; macular edema	One-time intravitreal injection	Taiwan Liposome Company	Phase 2	NCT03093701
SeeQ CdSe 655 Alt Nanoparticles (cadmium-selenium) NP	SeeQ Device	Retinitis Pigmentosa	Two intravitreal injections	2C Tech Corp	Phase 1	NCT04008771
Albumin-stabilized nanoparticle	Paclitaxel	Intraocular melanoma	Intravenous injections	Ohio State University Comprehensive Cancer Center	Phase 2	NCT00738361
EggPC liposomes	Latanoprost	Glaucoma	Subconjunctival injection	Singapore Eye Research Institute	Phase 1/2	NCT01987323
Pluronic^®^ F-127(PF) polymeric NP	Urea	Cataracts	Topical	Assiut University	Phase 2	NCT03001466
Ethylenediaminetetraacetic acid (EDTA) disodium salt and crocin liposomes	Hyaluronic acid	Meibomian gland dysfunction	Topical	University of Seville	Not applicable	NCT03617315
Nanoemulsion (OCU-310)	Brimonidine Tartrate	Meibomian gland dysfunction	Topical	Ocugen	Phase 3	NCT03785340
Sunitinib Malate (GB-102) MP	Aflibercept	Neovascular AMD	Intravitreal injection(s)	Graybug Vision	Phase 1	NCT03249740
(LAMELLEYE) Liposomal NP	Slecithin phospholipids, sphingomyelin and cholesterol, suspended in saline	Dry eye secondary to Sjögren Syndrome	Topical	NHS Greater Glasgow and Clyde	Not applicable	NCT03140111
AXR-159 ophthalmic solution (Micelles)	Integrins α4β1 and α4β7 antagonists	Dry eye	Topical	AxeroVision, Inc.	Phase 2	NCT03598699
KPI-121 (submicron suspension)	loteprednol etabonate	Ocular infections, irritations, and inflammation	Topical	Kala Pharmaceuticals, Inc.	Phase 3	NCT02163824
KPI-121 (submicron suspension)	loteprednol etabonate	Dry eye, keratoconjunctivitis sicca	Topical	Kala Pharmaceuticals, Inc.	Phase 3	NCT02813265
AR-1105	Dexamethasone	Macular edema due to retinal vein occlusion	Intravitreal implant	Aerie Pharmaceuticals	Phase 2	NCT03739593
AR-13503 implant	Aflibercept	Neovascular age-related macular degeneration	Intravitreal implant	Aerie Pharmaceuticals	Phase 1	NCT03835884
REMOGEN^®^ OMEGA (Microemulsion of polyunsaturated fatty acids and hydrating polymers)	Omega-3 fatty acids	Dry eye	Topical	TRB Chemedica AG	Not applicable	NCT02908282
Liposomes	Artificial tears	Dry eye	Spray	Aston University	Not applicable	NCT02420834
ENV 515	Travoprost	Glaucoma	Intracameral implant	Envisia Therapeutics	Phase 2	NCT02371746
OCS-01 (Cyclodextrin NP)	Dexamethasone	Corneal inflammation and post-operative pain	Topical	Oculis	Phase 2	NCT04130802

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
