# Peer review of "Ocular Drug Delivery: Advancements and Innovations"

_pharmaceutics, 2022, doi:10.3390/pharmaceutics14091931_

Round 1
Reviewer 1 Report
This is a timely and comprehensive review of different ocular drug delivery systems. It should be an interesting paper for researchers in the eye field because it gives beginners a good overview and established researchers a guidance when they want to decide which method used in their studies. The manuscript is well-written. I only have a few suggestions.
1. “There are more than 350 hereditary ocular diseases 94 including including retinitis pigmentosa, choroideremia, Stargardt disease, Leber’s 95 congenital amaurosis (LCA), glaucoma, and corneal dystrophies, involving a wide 96 diversity of genetic loci [6,7].”. Please remove glaucoma and corneal dystrophies since they are not hereditary disease although genetic link is one of the causes.
2. It would be helpful to have an illustration when describing the genome of AAV and rAAV
3. For each table, please modify the width of the column and font size so that the content of each column can fit into less than 3 lines. For example, Table 2 is misaligned in “Comments/ Mechanism”. The width of this column can be expanded so that the comments (outcomes) can fit into 1-3 lines. Now it is in many lines and difficult to read.
4. The illustration of drug administration routes to the anterior segment of the eye is very helpful for readers to understand the techniques. It would be helpful to have a similar illustration for drug delivery to the posterior segment.
Reviewer 2 Report
Review comments on pharmaceutics-1871102: Ocular Drug Delivery: Advancements and Innovations
The manuscript by Tian et al. reviewed recent advancements in ocular drug delivery. The authors put much effort into synthesizing the manuscript and summarizing data from previous studies and clinical trials. Unfortunately, the current manuscript has several limitations. The authors should consider the below comments to revise the manuscript.
1. The Abstract should be revised to clarify the coverage of this review. Please specify that this review covers both gene delivery and drug delivery. The Introduction section should be correspondingly revised.
2. The headings of sub-sections in this manuscript should be revised. They should be numbered following the journal guidelines (not using A, B, i, ii, iii). Currently, it contains section 2 (Ocular Gene Therapy), section 3 (Delivery system for ocular therapeutics), section 4 (Advancements in routes of administration), and section 4 (Conclusion). The Conclusion should be section 5. Section 2 has sub-sections A, B, and C. Sub-section A (Viral Vectors) starts with i (Adeno-associated viruses), and there are no other types of viruses. Sub-section B (Non-viral vectors) starts with ii (Organic Nanoparticles), and there is no Inorganic NPs. It is unclear why they were excluded (line 343).
3. Sub-section B of section 2 presents the delivery of drugs, not genes. It is unsuitable to put in section 2 - Ocular Gene Therapy. Are there any studies/ clinical trials employing organic NPs to deliver genes for ocular use?
4. Sub-section C in section 2 is about “Systemic delivery” via the oral route. Is it relevant to this review?
5. Section 3 - Delivery system for ocular therapeutics: this title seems to cover some drug delivery systems in section 2. It should be changed.
6. Table 2 has errors in arrangement and should be revised.
7. Table 4: there is a subcutaneous injection. Is it suitable for ocular delivery?
8. Lines 491-493: the sizes are better to present in nm.
9. References #108 – 130 were not cited in the main text.
Round 2
Reviewer 2 Report
The manuscript was considerably improved after a revision. There is only a minor point to revise as follows. In section 2.2, there was only one sub-section 2.2.1 (no sub-section 2.2.2). Thus, the authors should modify the section and sub-section numbering.
Author Response
Dear Reviewer,
Appreciate your effort reviewing our revised manuscript!
Thank you so much for your comments. According to the following comment, we have revised the title of 2.2 to "Organic nanoparticles (NPs)" and re-numbered the subsections accordingly. A revised manuscript has been submitted too.
The manuscript was considerably improved after a revision. There is only a minor point to revise as follows. In section 2.2, there was only one sub-section 2.2.1 (no sub-section 2.2.2). Thus, the authors should modify the section and sub-section numbering.
Best,
Haijiang Lin